# Geochemistry and Sources Apportionment of Major Ions and Dissolved Heavy Metals in a Small Watershed on the Tibetan Plateau

**Wencong Xing** [1,2]**, Lai Wei** [1,2]**, Wenmin Ma** [1,2]**, Jun Li** [1,*]**, Xiaolong Liu** [1,*]**, Jian Hu** [3] **and Xiaoxia Wang** [1,2]

1 Tianjin Key Laboratory of Water Resources and Environment, Tianjin Normal University, Tianjin 300387, China
2 School of Geography and Environmental Science, Tianjin Normal University, Tianjin 300387, China
3 Skate Key Laboratory of Urban and Regional Ecology, Research Center for Eco-Environmental Sciences, Chinese Academy of Sciences, Beijing 100085, China
* Correspondence: lijun5931@163.com (J.L.); xiaolong.liu@tjnu.edu.cn (X.L.)

**Abstract:** Due to environmental sensibility and fragility, the water chemistry revolution and heavy metals accumulation influenced by natural and anthropogenic processes in the rivers on the Tibetan Plateau have recently become a global concern. However, targeted studies in small watersheds on the Tibetan Plateau are relatively limited. A study of surface waters in Duilong Qu (DLQ), a small watershed located on the Tibetan Plateau, have been conducted to assess the impact of natural and anthropogenic activities on the water environment of the DLQ by analyzing the major ions and heavy metals (Cd, Cr, Mn, Fe, Ni, Cu, Zn, Pb, and As) in the river waters. The results of the analysis of major ions showed that $SO_4^{2-}$ and $HCO_3^-$ were the dominant anions and $Ca^{2+}$ was the dominant cation, indicating that the water chemistry of the river waters was mainly of the $HCO_3$-Ca type. The results of Piper diagram and Gibbs diagram analysis indicate that the water solute is mainly controlled by the weathering of carbonate rocks, followed by the influence of geothermal water confluence. Compared to the world river average, the concentrations of Cr, Pb, and As in the studied rivers were relatively high. The heavy metal concentrations satisfy the standards of WHO and GB (Chinese national standard) guidelines. The PCA-APCS-MLR model has been employed and evidenced as a reliable tool to identify the sources of the heavy metals in this study. The results revealed that the heavy metals in the DLQ are caused by natural sources, geothermal water, and mining operations. The primary sources of As (93.63%), Cr (93.07%), Mn (73.53%), Fe (59.54%), and Pb (58.28%) in the DLQ were geothermal water, while Zn (91.41%), Mn (20.67%), Fe (40.46%), and Pb (26.15%) originated mainly from natural sources. Additionally, Cu (91.41%) was primarily influenced by mining operations, and Ni originated from mining (53.61%) and geothermal water sources (46.39%), while Cd (97.88%) originated from unknown sources. In the high-flow season periods from 1992 to 2017, the As concentrations in the DLQ decreased significantly, which might result from increasing precipitation and runoff. Overall, the results of this study suggest that both natural and anthropogenic activities have jointly affected the solutes in small rivers on the Tibetan Plateau, and heavy metal pollution should be emphasized in the future.

**Keywords:** Duilong Qu; major ions; heavy metals; source appointment; PCA-APCS-MLR

## 1. Introduction

In the 21st century, many of the major problems facing humanity are related to water quality issues [1]. These problems will be exacerbated in the future by climate change due to increased water temperatures, melting glaciers, etc, and will further generate feedbacks to the deterioration of water environment globally. For example, according to a report developed by the Watershed Initiative, the water quality status of the Mississippi River basin was worsening because of agricultural activities, human sewage; and natural

works [2]; in China, industrialization, urban development, and intensive agriculture are the major drivers of water quality worsening [3]; in India, 70% of surface water and groundwater is contaminated with pollutants [4]. However, most of the current research has focused on rivers that are highly influenced by anthropogenic influences, while little study has been conducted to clarify the water quality evolution of rivers that are influenced by natural processes [2], especially on remote small river watersheds. Relevant studies are urgently needed for assessing global water quality evolution, and, further, to provide more information for water protection policy makers.

The state of river water around the world has been gradually changing, caused by changes in climate and human activities [5–7]. According to data provided by the World Health Organization (WHO) in 2019, one-third of the global population lacks access to safe drinking water [8]. The retreat of the Tibetan Plateau (TP) glaciers due to global warming have shown a remarkable influence on the chemical composition of water and the level of heavy metal pollution in TP rivers [5]. The TP is the source of 10 major rivers in Asia and is known as the "Water Tower of Asia" [9]. Since it is the supply of water for about 40% of the world's population, understanding the water chemistry and quality of TP rivers is essential for society and ecosystems [5]. The primary controlling mechanism of river water chemistry on the TP is natural processes, including rock weathering, soil erosions, and solute inputs from groundwater, which are also characterized by its geological background and regional water cycling [9]. Specifically, soluble substances originating from the soil have been demonstrated as a critical factor dominating the river water chemistry, especially for small watersheds [10,11]. Besides, detailed information about hydrochemical processes would be helpful in studying and understand the formation, migration, transformation, and enrichment of heavy metals in river waters. Studies have reported that heavy metals in TP rivers are influenced not only by rock weathering and water–rock interaction [12,13], but also by artificial activities, such as input of geothermal resources [14], discharge of domestic sewage [9], traffic discharge [15], and mining activities [16,17]. In fact, more and more studies show that the source of some water bodies pollution can only be attributed to natural geological processes [18–20]. Rock minerals (e.g., serpentine minerals, spinel, pyroxene, and olivine) contain a large amount of heavy metal elements [18]. Heavy metals in minerals are released during water–rock action, which, in turn, causes pollution of the water environment [19,20]. Therefore, it is crucial to identify the different sources of heavy metal contaminations and quantify the contribution of the sources to heavy metal contaminations in rivers on the TP.

Since the TP plays an essential role in water resources in Asia, previous studies have been devoted to the influence of climate change on the hydrology and water resources of the TP [21]. In contrast, there has been little discussion on water chemistry and the water environment in TP rivers related to climate change. The study found that more than 80% of the water ion balance in the rivers of the TP consists of $Ca^{2+}$, $Mg^{2+}$, $HCO_3^-$, and $SO_4^{2-}$ [9]. The primary controlling mechanisms of water chemistry are natural processes and exhibit apparent space heterogeneity [22]. For example, the water quality of rivers in the southern TP is dominated by the control of rock weathering [22], whereas the water quality of rivers in the north-central TP is also mainly influenced by evaporation-crystallization [9]. Moreover, the TP rivers are generally considered to be unpolluted due to sparse population and minimal industrial activity [5]. However, recent studies showed that concentrations of As, Pb, Cu, and Cd in southern TP rivers exceed the WHO drinking water guidelines due to anthropogenic processes (e.g., mining activities and sewage discharge) and geothermal water [9,23]. Thus far, however, the majority of previous research has concentrated on the major lakes and rivers of the TP, which often exhibit large runoffs [24,25]. Currently, there are relatively few studies on small watersheds, and the analysis of pollution sources is insufficient, although they are likely to play an essential role in the livelihood of residents. In fact, clarifying the characteristics and analyzing the heavy metal pollution sources in small watersheds not only enriches the water chemistry data of TP, but is also an important basis for accurate heavy metals contamination control in regional rivers. Therefore, the

research on heavy metals pollution source analysis in small watersheds should receive more attention.

Geothermal water is formed in the subsurface rock and has a heat source [26]. It has been found that the temperature of geothermal water can be as high as 289 °C [14]. Therefore, the discharge of geothermal water will cause the temperature of the surrounding rivers to rise and reduce the dissolved oxygen content (DOC) of the rivers, resulting in the deterioration of water quality. The high temperature and water–rock interaction make the geothermal water dissolve abundant anions and cations, with a high degree of mineralization [27]. Previous studies have confirmed that the geothermal water inflow affects the water quality of surrounding rivers [28]. For example, the concentrations of $K^+$, $Na^+$, $Cl^-$, and $SiO_2$ of the river water downstream in the Zaotang River (China) increased due to the inflow of hot spring water from the Rehai geothermal field [28]. Thorsten et al. [29] showed that concentrations of Mn, Zn, and Cu in Río Vacas Heladas (Chile) exceeded WHO guideline values for drinking water. Timperley et al. [30] estimated that 78% of the $Cl^-$ in the Waikato River (New Zealand) comes from geothermal fluids. More and more studies are proving the impact of geothermal water on rivers related to river runoff. The Büyük Menderes River (Turkey) runoff exceeded 26.651 $m^3/s$, and the emission of geothermal waters into the streams did not result in any environmental problems [31]. Blaine et al. [32] found that the fraction of the Yellowstone River (USA) from geothermal sources was about 0.2% during the high-flow season and 6.5% during the low-flow season. In addition, geothermal fluids contain very high concentrations of harmful chemical components, especially arsenic (As) [14]. Arsenic concentrations in water at El Tatio (Northern Chile) [33], Waikato River (New Zealand) [34], and Yellowstone National Park (USA) [35] were reported to be as high as 27,000 mg/L, 3800 mg/L, and 7800 mg/L, respectively. Thus, it was inferred that the discharge of geothermal water, which contains high As concentrations, might be a reason for the high As concentration and accumulation in the surrounding rivers. It is known that the long-term intake of high concentrations of As may induce endemic As poisoning [23]. Therefore, it is worth paying attention to the impact of geothermal water on the surrounding water quality. The TP is a region of intense geothermal activity in China [14]. The Lhasa River (LR) basin is situated in the mid-southern TP and is the main principal tributary of the middle Yarlung Tsangpo River [13]. As the core area of politics, economics, culture, transportation, and religion in the Tibet Autonomous Region, the LR basin is essential to Tibet's socioeconomic development [5,13]. The Duilong Qu (DLQ, Qu means river in the Tibetan language), a major tributary of the LR, is located downstream from the town of Yangbajing (YBJ) [12,23]. The YBJ geothermal field is the largest geothermal field on the TP, with the highest heat storage temperature (average temperature of 248 °C) in China [23]. Previous studies on YBJ geothermal water and surrounding areas have mainly focused on regional and geothermal geology [36]. However, it is still unclear how geothermal water discharge impacts the water quality of the DLQ. Thus, when the geothermal water and domestic sewage of surrounding residents flow into the DLQ [23], the water quality of the DLQ may be at risk of contamination [14,23]. As the water source for the production and living of residents [23], the water quality safety of the DLQ is a severe issue for the residents.

Most previous studies have focused on the evaluation of water quality by using methods such as the factor water quality identification index method and the water quality index (WQI) [37]. However, the investigation of pollution sources so far has been inadequate. Factor analysis (FA) and principal component analysis (PCA) have been used to identify the primary pollution sources in the environment [38]. However, the above methods have difficulty quantitatively describing the impact degree of major pollution sources [24]. The absolute principal component scores multiple linear regression model (APCS-MLR) based on PCA/FA has been proposed to quantify the contribution (in %) of identified pollutant sources [39]. The model has been effectively applied to identify pollution in air, soil, and surface water [38]. Previous studies have used the APCS-MLR model to quantify the contribution of pollution sources in rivers, showing that the analysis results are objective

and accurate [38]. In this study, the major ions and heavy metals of river water in the DLQ were investigated. The aims of the study are to: (1) analyze the major ionic composition and water quality of the DLQ; (2) identify the natural and/or anthropogenic sources of heavy metals; (3) study the effect of geothermal hot springs and climate change on As concentrations in river water in the DLQ. This study would reveal the effects of natural and anthropogenic activities on the solutes of small rivers on the TP and provide a scientific basis for protecting the water environment of river water in DLQ under the background of climate change.

## 2. Materials and Methods

### 2.1. Study Area

The DLQ ($29°38'$ N–$30°09'$ N, $90°31'$ E–$90°59'$ E) is a primary tributary in the middle reaches of the LR, originating from the southern foot of the Gangdese Mountain with a length of 137 km, a drainage area of 5093 km$^2$, and an elevation range of 3610–4130 m. The watershed is generally characterized by complex terrain with large portions of the region being deep valley. The DLQ valley plain stretches from northwest to southeast, with an average longitudinal gradient of about 5.63 ‰ and a riverbed width of 100~170 m. The climate of the DLQ valley plain is plateau temperate semi-arid monsoon, with an average annual temperature of 8 °C. The low-flow season is from November to April. About 89.1% of the yearly precipitation is primarily concentrated in the high-flow season (from June to September) [13]. The river runoff is mainly concentrated in summer, and the runoff from July to August accounts for 50% of the annual runoff. The annual mean runoff of the DLQ is 24 m$^3$/s, and the maximum peak runoff is 75 m$^3$/s [40]. The bedrock of the Lhasa valley area is dominated by Mesozoic limestone, metamorphic sandy slate, and Himalayan granite, in addition to Yanshanian granite, Quaternary clastic rocks, and eruptive rock.

### 2.2. Sampling and Analysis

The sampling protocol was based on the Technical Guidance on Water Quality Sampling (HJ 494-2009) and the geomorphological features of the watershed. In addition, we considered that the geothermal activity in Yangbajing (YBJ), the anthropogenic activity in the township, and the mining activity may cause changes in the water environment; therefore, we set up sampling points near these areas. A total of 23 surface water (depth < 0.5 m) samples were collected during the high-flow season (July–August) in 2015 (Figure 1), including 12 samples in the mainstream of the DLQ, 8 samples in tributaries of the DLQ, 2 samples in LR (two sampling sites are close to the confluence of two rivers and located at the upper stream and downstream of the confluence, respectively), and one geothermal water sample from the YBJ geothermal field.

The sampling protocol in this study follows a national standard method: the "Water Quality-Guidance on sampling techniques" (HJ 494-2009). Water samples used portable water samplers (Dongxiyi Co., Beijing, China) to collect surface water from where the water flow was most rapid to avoid the riparian effects, to ensure the water samples were representative. Pre-cleaned bottles were rinsed three times before being filled with water samples. Replicates and blank samples were collected and used to ensure the accuracy of the analysis and complete the determination. The pH, electrical conductivity (EC), water temperature (T), chlorophyll-a (Chl-a), and total dissolved solids (TDS) were measured in situ with a Multi-Parameter Water Quality Sonde (V2-4 6600, YSI, Yellow Springs, OH, USA) in the field. The collected water samples were filtered through disposable cellulose acetate membrane filters (0.45 μm) within 12 h of collection and then packed separately. The total alkalinity (Alk) was determined by the Gran titration method with 0.024 mol·L$^{-1}$ HCl in situ. Based on Alk, pH, T, and other anion and cation concentrations, the concentrations of $HCO_3^-$ and $CO_3^{2-}$ were calculated by PHREEQC, which was explored by USGS [41]. The concentrations of $Ca^{2+}$, $K^+$, $Mg^{2+}$, $Na^+$, and Si were measured by plasma atomic emitted spectrometer (ICP-OES, Optima 5300DV, PerkinElmer, Waltham, MA, USA) according to the current valid standard Water Quality-Determination of 32 Elements-Inductively

Coupled Plasma Emission Spectrometry (HJ776-2015). The standard curves were plotted with a standard solution of multiple elements (Agilent Technologies Inc., Santa Clara, CA, USA, Part number 6610030700) prepared on the same day. The concentrations of $Cl^-$, $NO_3^-$ and $SO_4^{2-}$ were measured using an ion chromatograph (Dionex ICS-2100, Dionex, Sunnyvale, CA, USA) according to the current valid standard Water Quality-Determination of Inorganic Anions ($F^-$, $Cl^-$, $NO_2^-$, $Br^-$, $NO_3^-$, $PO_4^{3-}$, $SO_3^{2-}$, and $SO_4^{2-}$) -Ion Chromatography (HJ84-2016). The working curves were plotted using the standard solution (National Center for Reference Materials, Product number GNM-M042193-2013). The analysis errors were within ±5%. The concentrations of heavy metals, i.e., cadmium (Cd), chromium (Cr), manganese (Mn), iron (Fe), nickel (Ni), copper (Cu), zinc (Zn), and lead (Pb), were quantified by inductively-coupled plasma mass spectrometry (ICP-MS, X-7series, Thermo Fisher Scientific, Cleveland, OH, USA) according to the recommended method by Environmental Quality Standards for Surface Water (GB3838-2002). The standard curves were plotted with a standard solution of multiple elements (Agilent Technologies Inc, Part number 8500-6940). The relative standard deviations of the results were better than 3%. The concentrations of arsenic (As) were quantified by an atomic fluorescence spectrophotometer (LC-AFS, AFS-830, Jitian Instrument Co., Beijing, China), of which the relative standard deviation was less than 1%. Standard solutions were purchased from the National Center for Reference Materials. Determination of high arsenic samples such as geothermal water by step dilution was done to the concentration range of the standard curves. The correlation coefficients of the standard curves of each item were obtained by linear regression (r) ≥ 0.9999, which is in accordance with the requirements of Water Environment Monitoring Specification (SL219-2013). Replicates were also measured with water samples to check the stability. In addition, the standard solution corresponding to the measured sample content was added for quality control. All the precisions of measurements of the replicates and samples were better than ±3%. The acid used for the analysis was of high purity and the water was of high purity. Each batch of samples was blanked throughout the experiment to eliminate any contamination during sample processing and determination.

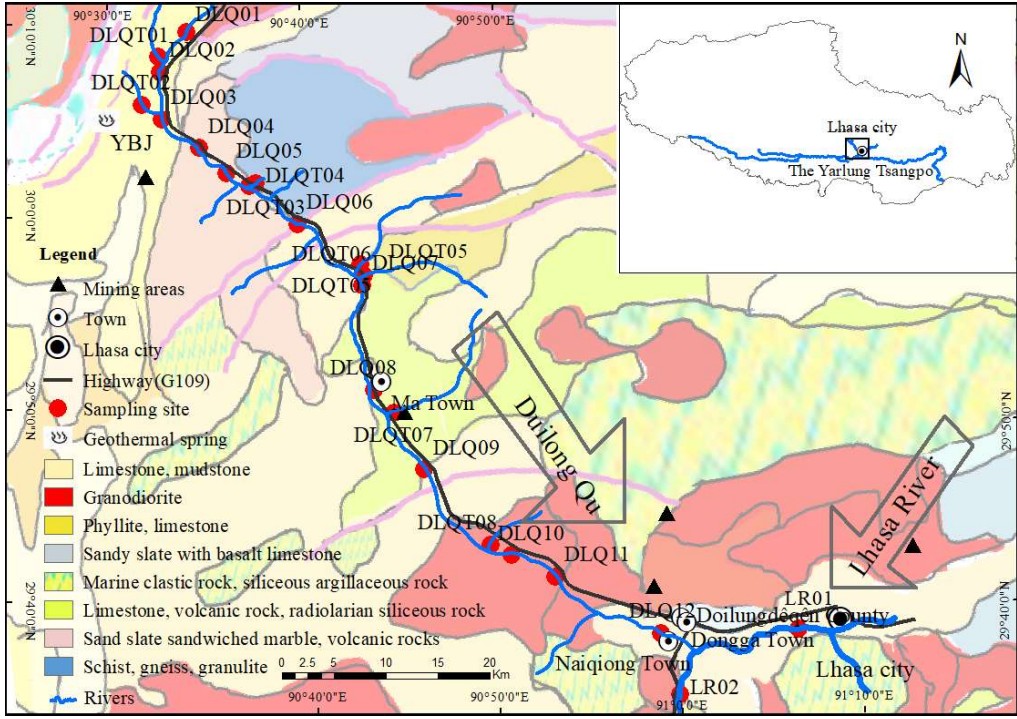

**Figure 1.** Distribution of Sampling sites in DLQ.

### 2.3. Analytical Methods

Sources of heavy metals in surface waters of the LR and DLQ were analyzed using correlation analysis, principal component analysis (PCA), and the principal component analysis-absolute principal component score-multiple linear regression receptor model (PCA-APCS-MLR). The APCS-MLR was performed for quantitative source assignment using the normalized factor scores and eigenvectors acquired by PCA. The main steps of the PCA-APCS-MLR are as follows [39]:

Step 1: All heavy metal concentration data are normalized using Equation (1):

$$Z_{ij} = (C_{ij} - \overline{C_i})/\sigma_i \tag{1}$$

where $C_{ij}$ is the accumulation value of heavy metal $i$ at location $j$, and $C_i$ and $\sigma_i$ are the mean concentration and the standard deviation for heavy metal $i$, respectively.

Step 2: The true zero for calculating the scores of individual factors is obtained by introducing an artificial sample with zero concentration of all variables by the formula:

$$(Z_0)_i = (0 - \overline{C_i})/\sigma_i \tag{2}$$

where $(Z_0)_i$ is established as the absolute zero concentration.

Step 3: The PCA approach can be used to obtain factor scores for variables from normalized heavy metals concentration data. The factor score of $(Z_0)_i$ was subtracted from the factor score of $Z_{ij}$ to estimate the APCS for each heavy metal.

Step 4: The $C_i$ of the source contribution was obtained with a multiple linear regression procedure:

$$C_i = b_{0i} + \sum_{p=1}^{n} (b_{pi} \times \mathrm{APCS}_p) \tag{3}$$

where $C_i$ is the estimated value of heavy metal concentration, $b_{0i}$ is the constant term of multiple regressions, $b_{pi}$ is the regression coefficient of the multivariate linearity, and the mean of $b_{pi} \times \mathrm{APCS}_p$ is the contribution value of the source $p$.

## 3. Results and Discussion

### 3.1. Physicochemical Properties

Table 1 shows the physicochemical properties of the rivers. The pH value is a common indicator of natural water acidity, which can affect the form and activity of heavy metals [9,13]. The pH value of YBJ was 9.5, which was higher than that of natural water (6–9) [13], indicating that the effects of the deep thermal water were dominated by sodium chlorite fluid [42]. The pH values of river water in the LR and DLQ ranged from 7.0–8.4, indicating a strong buffering ability of the river water in the context of the inputs of alkaline geothermal water from the YBJ geothermal field [12,13]. In addition, heavy metals are more easily deposited in sediments in alkaline water environments. Thus, the actual concentrations of heavy metals in rivers may be underestimated [9]. The high temperature of YBJ geothermal water was up to 80 °C. However, the temperatures in the DLQ (20.0–21.1 °C) were not increased significantly. The highest concentration of Chl-a (23.9 µg·L$^{-1}$) was observed at the sampling site near Naiqiong town, which may be related to the bloom of algae in summer [43]. Naiqiong Town is well-developed in agriculture, and agricultural activities such as fertilization and planting can increase the accumulation of nutrients in rivers [10,43,44]. In addition, high precipitation and high runoff during the high-flow season may transport more nutrients to a river [45,46], contributing to the high Chl-a concentration in the river near Naiqiong town. The DO concentrations of river waters in the LR and DLQ ranged from 6.3 to 6.6 mg·L$^{-1}$ (DO saturation ranged from 94.7% to 100.1%). The DO concentration of river water in the DLQ was higher when compared with rivers at the same altitude on the TP, such as the Requ River (6.0 mg·L$^{-1}$) and the Chumaer River (5.6 mg·L$^{-1}$), which may be caused by the accelerated mixing of water and oxygen in ambient air due to the fast flow rate of the river water in the DLQ [47]. The EC value of the water in YBJ was 2.9 mS·cm$^{-1}$, and the TDS was 1422 mg·L$^{-1}$, which belonged to brackish

water (1–3 g·L$^{-1}$) [48]. The EC values of the LR and DLQ rivers ranged between 0.04 and 0.2 mS·cm$^{-1}$, and the TDS values of those rivers ranged from 33 to 150 mg·L$^{-1}$. The flow rate in the DLQ during the high-flow season can reach 75.1 m$^3$/s, while the geothermal water discharge is 0.389 m$^3$/s [49]. Therefore, YBJ geothermal water discharge has little effect on the solute level of the mainstream during the high-flow season, and the DLQ still exhibits typical freshwater characteristics (<1 g·L$^{-1}$) [13]. Compared with several large rivers on the TP, such as the Mekong River and the Yarlung Tsangpo River [5], the lower TDS and EC values in the DLQ rivers indicate that the influence of alkaline thermal springs on the hydrochemical parameters of the river water is limited.

**Table 1.** Physical-chemical parameters of surface water in the DLQ, LR, and YBJ geothermal water.

| Sites | | pH | T (°C) | DO (mg·L$^{-1}$) | Chl-a (µg·L$^{-1}$) | EC (mS·cm$^{-1}$) | TDS (mg·L$^{-1}$) | Reference |
|---|---|---|---|---|---|---|---|---|
| YBJ | | 9.5 | 80 | | | 2.9 | 1422 | |
| LR 01 | | 7.5 | 20.4 | 6.4 | 2.2 | 0.1 | 83 | |
| LR 02 | | 7.1 | 20.7 | 6.3 | 2.6 | 0.2 | 106 | This study |
| DLQ | Min | 7.0 | 20.0 | 6.3 | 0.5 | 0.04 | 33 | |
| (*n* = 20) | Mean | 7.6 | 20.5 | 6.4 | 10.1 | 0.1 | 84.5 | |
| | Max | 8.4 | 21.1 | 6.6 | 23.9 | 0.2 | 150 | |
| Requ River (Source region of the Yellow River) | Mean | / | / | 6.0 | / | 0.2 | 417 | [50] |
| Chumaer River (Source region of the Yangtze River) | Mean | 8.5 | 19.46 | 5.6 | / | 0.2 | 1884 | [51] |
| Mekong River | Mean | / | / | / | / | 0.3 | 302 | [5] |
| Yarlung Tsangpo River | Mean | / | / | / | / | 0.1 | 112 | |

Note(s): "/" means no detection.

### 3.2. Main Ionic Components and Hydrochemical Types

Table 2 shows the major ion concentrations in the study area. The major ions in YBJ geothermal water and rivers are markedly different, which is related to the difference in geological conditions. The concentrations of ions in YBJ geothermal water are higher than those in the rivers (LR and DLQ).

**Table 2.** Statistics of major ion concentrations in the DLQ, LR, and YBJ geothermal water (mg/L).

| Sites | | K$^+$ | Na$^+$ | Ca$^{2+}$ | Mg$^{2+}$ | Cl$^-$ | SO$_4^{2-}$ | HCO$_3^-$ | NO$_3^-$ | Si |
|---|---|---|---|---|---|---|---|---|---|---|
| YBJ | | 137.2 | 376.0 | 4.3 | 0.4 | 440.7 | 71.8 | 357.2 | 0.4 | 237.5 |
| LR 01 | | 1.4 | 6.8 | 31.25 | 5.5 | 14.4 | 21.6 | 86.5 | 0.6 | 3.2 |
| LR 02 | | 1.4 | 4.9 | 19.39 | 1.6 | 7.3 | 12.1 | 54.6 | 1.8 | 3.2 |
| DLQ | Min | 0.7 | 1.3 | 6.2 | 0.3 | 1.2 | 1.5 | 20.4 | 0.2 | 1.8 |
| (*n* = 20) | Mean | 1.6 | 4.2 | 18.6 | 1.66 | 4.3 | 16.1 | 41.5 | 1.2 | 3.0 |
| | Max | 4.5 | 8.2 | 33.0 | 2.8 | 13.4 | 52.1 | 75.8 | 3.2 | 4.7 |

The dominant anion and cation in YBJ geothermal water were Na$^+$ and Cl$^-$, respectively (Table 2), and the water chemical type was Cl-Na type (Figure 2). This might be because the alkaline solutions are prone to dissolving silicates, and the interaction of alkaline water with rock minerals (mainly granite in the deep layer of the YBJ geothermal field) lead to an increase in the concentrations of SiO$_2$, Na$^+$, and K$^+$ [26]. Ca$^{2+}$ accounted for the highest total cationic equivalent charge of 63~87% in the river water of the DLQ, followed by Na$^+$+ K$^+$ (4–26%) and Mg$^{2+}$ (3–9%), while HCO$_3^-$ (22–84%) was the dominant anion, followed by SO$_4^{2-}$ (6–75%) and Cl$^-$ (2–19%). Therefore, 78% of the water samples in the DLQ belong to the HCO$_3^-$-Ca types, and 22% of the water samples are SO$_4$·Cl-Ca types. The water samples in the DLQ near the YBJ geothermal water belong to the SO$_4$·Cl-Ca type, indicating that the contribution of solutes in the geothermal field cannot be ignored. Previous studies have shown that geothermal water contains high concentrations of Cl$^-$ [27,32]. In this study, the Cl$^-$ concentration in YBJ was as high as 440.7 mg/L (Table 2). In addition, the highest concentrations of SO$_4^{2-}$ (52.1 mg/L) were observed near the YBJ geothermal water, which is mainly due to the fact that the H$_2$S and SO$_2$ contained in the geothermal water are easily oxidized to form SO$_4^{2-}$ during the upward migration process [27,52]. The water

chemistry of the DLQ gradually changed from the $SO_4 \cdot Cl$-Ca type to the $HCO_3$-Ca type with the direction of water flow, mainly due to more dissolved carbonate flowing into the river and the dilution of $SO_4^{2-}$ and $Cl^-$ in the river. $Ca^{2+}$ was the dominant cation of river water in LR, accounting for 69–71% of the cations, while $HCO_3^-$ was the dominant anion, accounting for 71–74% of the anions. Thus, the water chemistry type of LR is $HCO_3$-Ca, which is consistent with previous studies [5,22], indicating the important contribution of carbonate weathering to the ionic composition of river water in LR [13].

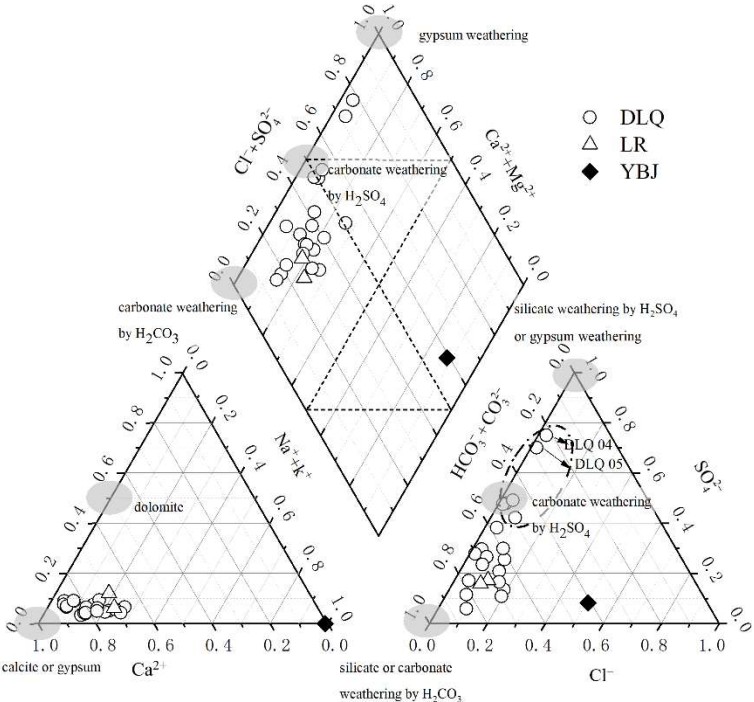

**Figure 2.** Piper diagram showing major ion compositions of the river waters. The end-members were taken from Spence and Telmer [53] and Li et al. [48].

On the Gibbs diagram (Figure 3), the scatter of major ions in the water of YBJ was outside the Gibbs diagram, which was mainly due to the high $Na^+/(Na^+ + Ca^{2+})$ weight ratio because of the high $Na^+$ concentrations (376.0 mg/L) and low concentrations of $Mg^{2+}$ (0.4 mg/L) and $Ca^{2+}$ (4.3 mg/L) in the geothermal water [27]. The $Na^+/(Na^+ + Ca^{2+})$ ratio in the LR and DLQ was less than 0.5, and the TDS concentration was moderate, being mainly controlled by rock weathering [54]. Similar findings were also found by Qin et al. [13] in the LR watershed. In the mixing diagram (Figure 2), most of the water samples in the LR and DLQ were located near the carbonate endmember, indicating that carbonate weathering driven by carbonic acid and sulfuric acid has a significant impact on the major ion chemistry of the LR and DLQ. Generally, the dissolution and weathering of carbonate rocks (calcite, dolomite, etc.) in natural water is the main source of $Ca^{2+}$ and $HCO_3^-$ [5,9,55], and the weathering of magnesium-bearing carbonate rocks also provide $Mg^{2+}$ in the river [55]. In this study, $Ca^{2+}$, $Mg^{2+}$, and $HCO_3^-$ showed a highly significant correlation ($p < 0.01$), which also proved that these ions mainly originated from the weathering of carbonate rocks [13,55]. The geology of the LR basin contains a higher proportion of Mesozoic limestone, a major weathering source of Ca, Mg, and $HCO_3^-$ [13]. Furthermore, $HCO_3^-$ showed positive correlation with $Ca^{2+}$ ($R^2 = 0.82$, $p < 0.01$) and $Mg^{2+}$ ($R^2 = 0.75$, $p < 0.01$), suggesting that dissolved calcite was the most significant contribution to river ions, followed by dolomite [55]. Basically, $Cl^-$ is primarily generated from the dissolution of evaporites [5,9]. $K^+$ and $Na^+$ are mainly derived from the dissolution of evaporites and the weathering of silicate minerals [9,56]. $Cl^-$, $K^+$, and $Na^+$ were highly significantly correlated ($p < 0.01$), indicating that these elements had similar sources. In addition, $Na^+$ was found to be related to Si ($R^2 = 0.51$, $p < 0.05$). Therefore, $Cl^-$, $K^+$, $Na^+$, and Si were generated

primarily by the dissolution of evaporites and/or the weathering of silicate minerals [9,55]. The source of ions in the river water in the DLQ is primarily influenced by rock weathering and dissolution.

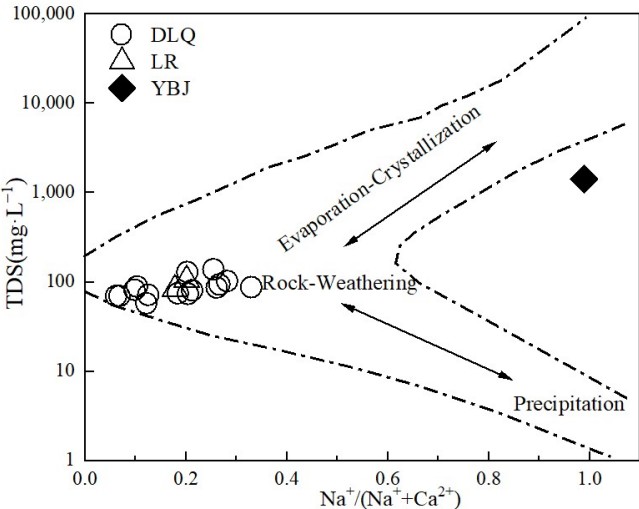

**Figure 3.** The Gibbs graph of major ion compositions in DLQ, LR, and YBJ geothermal water.

Long-term interactions between geothermal fluids and host rocks lead to As accumulation in hydrothermal systems [57]. As mentioned before, high As concentrations have been found in rivers around the world [33]. Thus, arsenic is often regarded as a characteristic element of geothermal water [23,52]. In this study, a significantly positive association was found between $SO_4^{2-}$ and As ($R^2 = 0.67$, $p < 0.01$), indicating that $SO_4^{2-}$ might have similar origins to As, and which can be attributed to geothermal water. In addition, oxidative dissolution of As-containing sulphides may also result in $SO_4^{2-}$ and As enrichment [58]. Moreover, $SO_4^{2-}$ showed a significantly positive correlation with $Ca^{2+}$ ($R^2 = 0.70$, $p < 0.01$). This could be due to: (1) the dissolution of gypsum [5]; (2) the erosion of the carbonate rocks driven by the sulfuric acid, which was produced from the oxidation of $H_2S$ in geothermal water [27]. However, gypsum is rarely distributed in the DLQ watershed; therefore, the contribution of gypsum dissolution is negligible [59]. Generally, $NO_3^-$ is an important water chemistry parameter, which mainly originates from agricultural fertilizers, soil organic nitrogen, and domestic sewage discharge [9,55,60]. Previous studies reported that Lhasa city has 182,000 inhabitants [5], and the ecology of river waters in the LR is damaged by domestic, industrial, and agricultural wastewater discharged into the river [9,13]. Hence, slightly higher $NO_3^-$ concentrations at sampling sites around Lhasa city indicate the influence of urban sewage and agricultural activities on the LR (Table 2) [13].

### 3.3. Exposure Level of Heavy Metals and Assessment of Water Quality

Since these rivers are the major drinking water sources for residents, the water quality of the study area was evaluated. The criteria used for evaluation include the Environmental Quality Standard for Surface Water (GB3838-2002), Sanitation Standard for Drinking Water (GB5749-2006), World Health Organization (WHO), National Primary Drinking Water Regulations (NPDWR), and National Secondary Drinking Water Regulations (NSDWR) (Table 3).

**Table 3.** Descriptive statistical results of heavy metals in surface water (mean $\pm$ standard deviation, µg/L).

| | Cd | Cr | Mn | Fe | Ni | Cu | Zn | Pb | As | Reference |
|---|---|---|---|---|---|---|---|---|---|---|
| YBJ | 0.10 | 1.53 | 38.85 | 358.79 | 2.15 | 8.13 | 17.94 | 4.88 | 3532.21 | |
| LR 01 | 0.04 | 1.45 | 5.19 | 39.27 | 0.75 | 2.89 | 4.20 | 0.32 | 3.06 | This study |
| LR 02 | 0.05 | 1.99 | 13.43 | 47.19 | 1.10 | 0.91 | 2.44 | 0.50 | 5.05 | |
| DLQ (n = 20) | 0.05 ± 0.01 | 1.65 ± 0.42 | 7.30 ± 3.65 | 78.36 ± 40.93 | 0.84 ± 0.4 | 1.03 ± 0.62 | 5.26 ± 3.58 | 0.53 ± 0.27 | 3.33 ± 2.98 | |
| Source region of the Yangtze River | 0.02 | 1.83 | / | / | 1.73 | 5.87 | 3.52 | 1.06 | 10.02 | [16] |
| Natural waters in Tibet | | 4.13 | 0.77 | 95.57 | 1.01 | 0.38 | 4.91 | 0.03 | 6.11 | [61] |
| World average | 0.08 | 0.7 | 34 | 66 | 0.8 | 1.48 | 10 | 0.03 | 0.62 | [62] |
| WHO | 3 | 50 (P) | 400 (C) | 3000 | 70 | 2000 | 3000 | 10 | 10 | [63] |
| GB I/V | 1 | 10 | / | / | 20 | 10 | 50 | 10 | 50/100 | [64] |
| GB5749-2006 | 5 | 50 | 100 | 300 | / | 1000 | 1000 | 10 | 50 | [65] |
| NPDWR | 5 | 100 | / | / | / | / | / | 15 | 10 | [66] |
| NSDWR | - | 50 | 300 | / | 1000 | 5000 | / | 100 | / | |

Note(s): "/" means no detection. P: provisional guideline value because of uncertainties in the health database; C: concentrations of the substance at or below the health-based guideline value may affect the appearance, taste, or odor of the water, leading to consumer complaints.

The results show that heavy metal concentrations in YBJ geothermal water meet the water quality standards (or guidelines), except for As and Fe. The concentration of Fe in YBJ geothermal water was 358.79 mg/L, which exceeded the GB standard for drinking water (300 mg/L). The As concentration in YBJ geothermal water (3532.21 mg/L) was much higher than the limit value of class V of the GB standard level for Surface Water (100 g/L). Since high concentrations of As were found in YBJ geothermal water, it can be said that the influx of geothermal water impacts river water quality [14]. A previous study found that the discharge of YBJ geothermal water resulted in As contamination in the DLQ during the low-flow season [12]. However, the concentrations of heavy metals of river waters in the LR and DLQ meet the water quality standards (or guidelines) in this study, indicating a weak impact of geothermal water on the water quality of river waters in the LR and DLQ, due to the strong dilution effect of the river during the high-flow season [12,25]. This is similar to the results of heavy metals during the high-flow season of DLQ by Li et al. [23]. Moreover, it is noted that the concentrations of Cr, Pb, and As in the rivers of the LR and DLQ are significantly above average compared to other rivers [62]. The concentrations of Mn, Pb, and Cu in river waters in the LR and DLQ were much higher than other natural waters in Tibet (0.77 mg/L for Mn, 0.03 mg/L for Pb, and 0.38 mg/L for Cu) [61]. The concentrations of heavy metals in river waters in the LR and DLQ were comparable with those in river water of the source area in the Yangtze River [16], except for Cd. In addition to rock weathering, geothermal water inflow, and discharge of urban sewage and agricultural wastewater [12,13,24], mining activities increase heavy metal concentrations in the river waters in the DLQ and LR due to the abundant mineral deposits (e.g., Cu and Pb-Zn mines) in the LR watershed [17,41]. The results showed that the confluence of geothermal water has not led to heavy metal contamination in the studied river waters. However, it is noteworthy that the occurrence of sudden geological disasters may cause abruptly high concentrations of heavy metals in rivers in a short period of time, especially for As. Therefore, it is important to develop geothermal water resources rationally while minimizing or avoiding the hazards caused by unexpected events. This will better ensure the sustainable use of geothermal water resources.

### 3.4. Spatio-Temporal Distribution of Heavy Metals

3.4.1. Spatio Distribution of Heavy Metals

The spatial distribution of different heavy metals of river water in DLQ demonstrated that (Figure 4), except for the YBJ geothermal water, the concentrations of heavy metals in other Sampling sites are generally low (Table 3). Overall, the concentrations of As, Fe, Mn, Zn, Pb, and Cu of river water in DLQ increase after the inflow of YBJ geothermal water, indicating the contribution of geothermal water, while the decrease, afterward, was mainly due to the water self-purification and the inflowing of the tributaries. In addition, the concentration of heavy metals (except As) of river water in DLQ increased again after passing near the towns of Dongga and Naiqiong (DLQ12), which reflects the influence of sewage input from the towns.

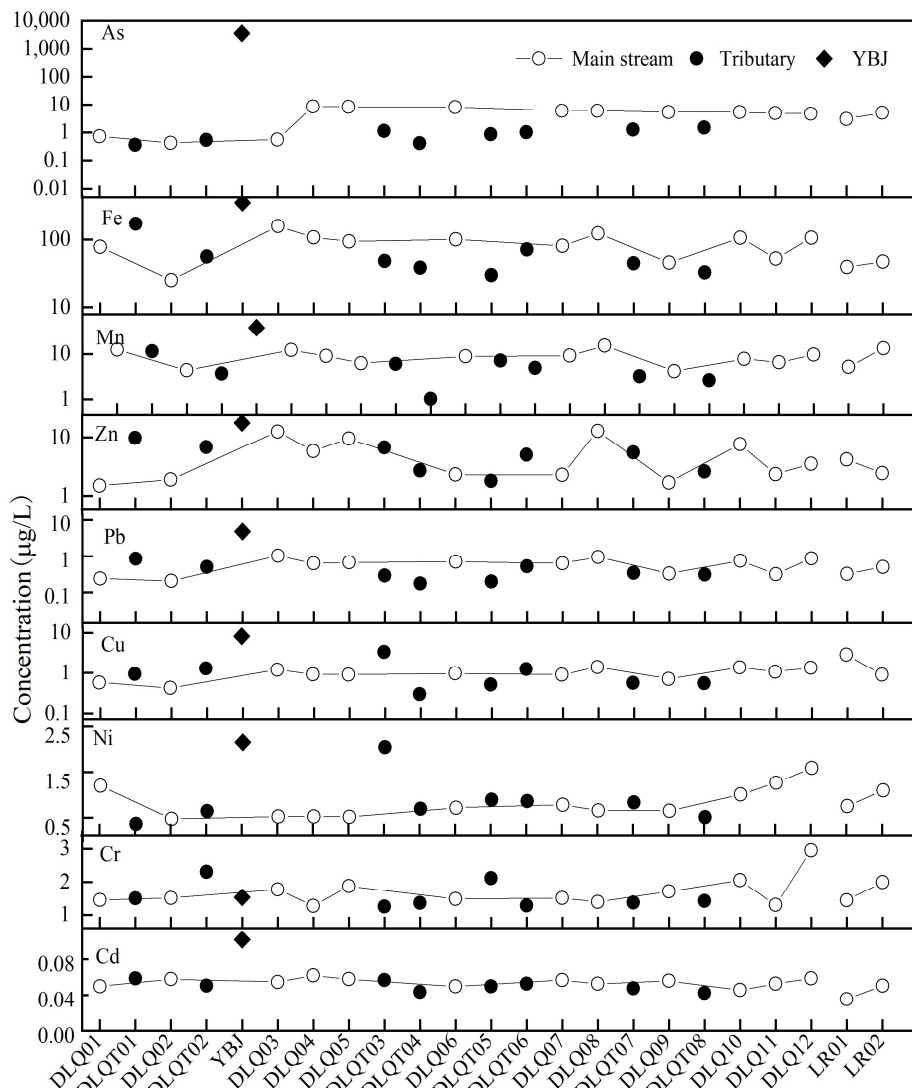

**Figure 4.** Spatial variations in heavy metals along the flow direction in DLQ.

The concentration of As gradually decreased from DLQ04 to DLQ12, indicating that it may be influenced by river water dilution and sediment adsorption in the riverbed [12]. The flow rate of the mainstream in DLQ can reach 75.1 m$^3$/s during the high-flow season [40], and concentrations of As in the tributaries stream of DLQ are generally lower than those in the mainstream (Figure 4). Hence, the decrease in As concentration was mainly attributed to the dilution effect of runoff [12]. In addition, adsorption from streambed sediments, especially by iron oxyhydroxides [58], may play an important role in reducing arsenic concentrations in the river [14]. A study in May 2007 showed that most of the As in rivers could be removed from the river by adsorption by river sediments [57]. However, the runoff rate in July and August (73.6–75.1 m$^3$/s) of DLQ is much larger than the runoff rate in May (10.1 m$^3$/s). Therefore, sediment sorption is limited due to increased runoff rates, resulting in the desorption of As from the sediment [57]. The curves of Fe, Mn, Zn, and Pb were very similar (Figure 4), indicating that familiar sources exist between them [12]. High concentrations of Fe (123.61 mg/L), Mn (12.28 mg/L), Zn (12.96 mg/L), and Pb (0.95 mg/L) were observed in DLQ08, which is related to the proximity of the sampling site to agricultural and grazing areas (Ma Town). Livestock grazing in agricultural and pastoral regions can lead to the release of metal elements in the surrounding shallow water sediments [12,13]. In addition, Wang et al. [15] found that transportation activities caused elevated Zn and Pb concentrations in soils on the highway G109 on the Qinghai-Tibet Plateau and affected the

area beyond 100 m. Nevertheless, Highway G109 is less than 100 m away from some DLQ sampling sites, indicating that Zn and Pb may be influenced by traffic activities. The highest concentrations of Cu (3.31 mg/L) and Ni (2.04 mg/L) existed in DLQT03, indicating the influence of mining activities in the surrounding metal deposits [13,17]. However, this was not observed in the mainstream, indicating a strong dilution effect of surface runoff [13]. This study has shown that the heavy metals in fertilizers are mainly Cr and Co, and agricultural activities can lead to elevated Cr concentrations in the river [25]. In this study, a high Cr concentration (2.96 mg/L) was observed in the site near the town of Naiqiong, where agriculture has been well-developed, and this was associated with the heavy use of pesticides and fertilizers. Li et al. [25], likewise, observed highly concentrated Cr in the Nianchu River, a grain producing region in Tibet. Except for Zn and Cu, the heavy metal concentrations of LR02 were higher than those of LR01, indicating that the influx of DLQ had an impact on the water quality of LR. It is worth noting that the concentrations of As (5.05 mg/L) and Mn (13.43 mg/L) of LR02 were higher than those of DLQ12 (4.75 mg/L for As and 9.65 mg/L for Mn), indicating that LR is also affected by mining activities, domestic waste, and municipal sewage [13,23,25]. In summary, the spatial distribution of heavy metals in the LR and DLQ was closely related to geothermal resources and human activities.

### 3.4.2. Impact of Climate Change on As Concentration in DLQ

Since the 1980s, the LR basin has been affected by global warming, and the annual average temperature and precipitation in the basin have been increasing [25,67]. Since the precipitation and flow data of DLQ are lacking, the meteorological and hydrological data of Lhasa city were selected as a reference in this study. The annual precipitation in Lhasa increased at a rate of 28.81 mm/10 a, and the summer precipitation increased at 32.18 mm/10 a from 1980 to 2019 [68]. Although the hydrological data of Lhasa city cannot directly represent the precipitation conditions of DLQ, they still have some reference values for the city's precipitation variation trend. The trend of the average river runoff in DLQ is consistent with precipitation since precipitation is the primary source of recharge in the DLQ river's precipitation [23,68]. In addition, small-scale glaciers have developed at the headwaters of the DLQ tributaries, and meltwater from the ice and snow due to climate warming will also cause an increase in river runoff [69].

The seasonal and spatial variation in different trace metals differed because of their complex governing factors [12]. Unlike other elements, As concentrations in river water affected by geothermal activity exhibit significant seasonal variation [12]. Li et al. [23] observed that the As level of the LR during the low-flow season was about three times higher than during the high-flow season. This is due to the more significant contribution of geothermal water during the low-flow season and the stronger dilution process during the high-flow season [25,70]. Thus, the As level is highly dependent on the river's hydrological state [23]. Some studies have shown that, although As levels in rivers are also affected by pH and adsorption/desorption processes [52], changes in As concentrations are mainly influenced by river runoff [23]. Therefore, we collected the As data in river water of the DLQ from 1992 to 1994 [40], 1997 to 1999 [49], 2010 [23], and 2017 [71], respectively, to analyze the effects of precipitation and runoff changes on As (Figure 5).

From 1992 to 2017, the As concentrations in the river showed a decreasing trend (35.5 g/L~1.2 g/L) during the high-flow season, which may be related to a significant increase in precipitation and runoff due to climate change. In addition, from 1992 to 2010, the most significant reduction was found in the high-flow season (35.5 g/L~12.9 g/L) compared to the normal-flow season (59.9 g/L~40 g/L) and the low-flow season (292 g/L~205.6 g/L), which was primarily caused by the stronger dilution of the river in the high-flow season [25]. Therefore, based on the variations in As concentrations in this study, it can be inferred to some extent that the recent increase in precipitation and runoff in LR caused by climate change may lead to a decrease in the concentration of heavy metals in river water. Similarly, metal concentrations in the Rhine River were diluted by the river due to climate change as observed by Zwolsman et al. [72].

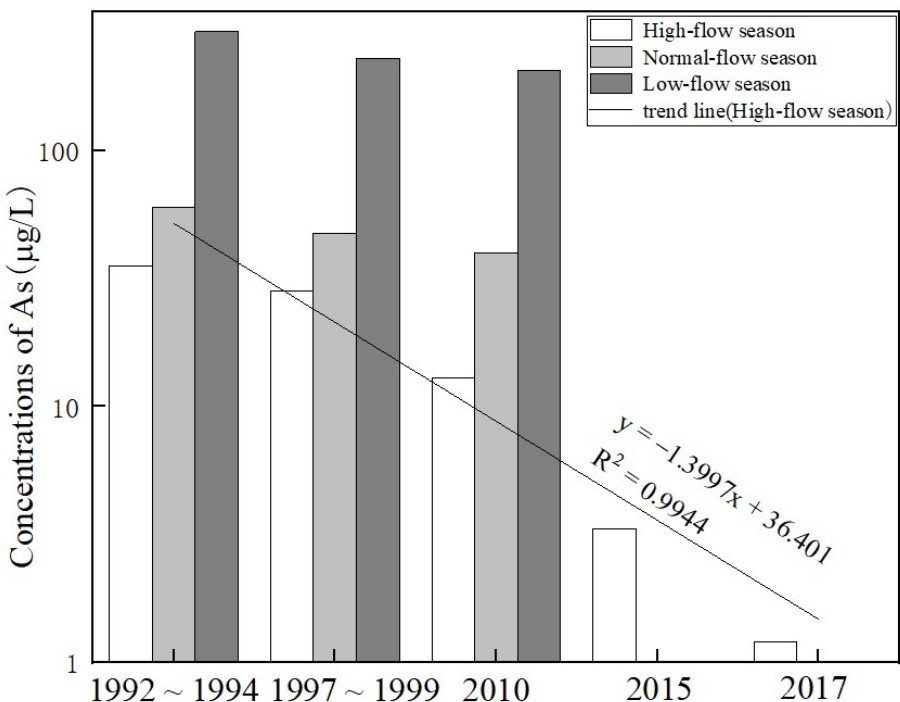

**Figure 5.** The seasonal variations in As in DLQ during the period from 1992 to 2017. Data were referred to [23,40,49,71].

### 3.5. Pollution Source Identification Based on the Pearson Correction, PCA and PCA-MLR

#### 3.5.1. Correlation Analysis

The results of Pearson correlation coefficients between the measured heavy metals are listed in Table 4. A positive correlation was found among Mn-Fe ($p < 0.01$, $R^2 = 0.67$), Mn-Pb ($p < 0.01$, $R^2 = 0.64$), Fe-Pb ($p < 0.01$, $R^2 = 0.90$), Pb-Zn ($p < 0.01$, $R^2 = 0.72$), and Fe-Zn ($p < 0.01$, $R^2 = 0.71$), suggesting a common source of these heavy metals. Fe and Mn are mainly derived from the crust [12,25]. Therefore, Mn, Fe, Pb, and Zn are mainly derived from a natural geological background. Moreover, a significant positive correlation was observed between Cd and Fe ($p < 0.01$, $R^2 = 0.43$), which may be related to the migration and transformation between them, because they were not similar in terms of spatial distribution pattern (Figure 4). The abundance of Cu in the LR basin indicates that Cu is mainly derived from mining activities [12,25]. Ni showed positive correlations with Cu ($p < 0.05$, $R^2 = 0.52$), K$^+$ ($p < 0.05$, $R^2 = 0.48$), and Na$^+$ ($p < 0.05$, $R^2 = 0.47$) in the DLQ, indicating that Ni is influenced by mining activities [25] and natural processes [24]. Cr and As had no significant correlation with other heavy metals, indicating that they are not controlled by a single factor in the study area [12]. Arsenic showed a significant association with NO$_3^-$ ($p < 0.01$, $R^2 = 0.58$), suggesting that As might be related to geothermal activity, agricultural operations, and grazing [12,25].

#### 3.5.2. Principal Component Analysis

PCA was performed for nine heavy metals (Table 5), with a KMO value of 0.561 (> 0.5) and Bartlet's sphericity test result of 0.000 (0.000 < 0.05), showing that the heavy metals were appropriate for PCA analysis. Three principal components (PCs) with eigenvalues greater than one were extracted, the cumulative variance contribution was 57.85%, and control factors for each PC were explained as follows.

**Table 4.** The Pearson correlation analysis of heavy metals in DLQ.

| | Cd | Cr | Mn | Fe | Ni | Cu | Zn | Pb | As | Ca$^{2+}$ | K$^{+}$ | Mg$^{2+}$ | Na$^{+}$ | Si | Cl$^{-}$ | NO$_3^{-}$ | SO$_4^{2-}$ |
|---|---|---|---|---|---|---|---|---|---|---|---|---|---|---|---|---|---|
| Cr | 0.12 | | | | | | | | | | | | | | | | |
| Mn | 0.34 | 0.14 | | | | | | | | | | | | | | | |
| Fe | 0.43 * | 0.11 | 0.67 ** | | | | | | | | | | | | | | |
| Ni | 0.06 | 0.18 | 0.09 | −0.21 | | | | | | | | | | | | | |
| Cu | −0.12 | −0.06 | 0.07 | 0.04 | 0.52 * | | | | | | | | | | | | |
| Zn | 0.22 | 0.00 | 0.39 | 0.71 ** | −0.22 | 0.29 | | | | | | | | | | | |
| Pb | 0.39 | 0.31 | 0.64 ** | 0.90 ** | −0.18 | 0.11 | 0.72 ** | | | | | | | | | | |
| As | 0.26 | 0.07 | 0.29 | 0.24 | −0.05 | 0.02 | 0.04 | 0.39 | | | | | | | | | |
| Ca$^{2+}$ | −0.21 | −0.20 | 0.18 | −0.09 | 0.00 | 0.17 | −0.02 | 0.03 | 0.52 * | | | | | | | | |
| K$^{+}$ | −0.04 | 0.52 * | 0.17 | −0.06 | 0.48 * | −0.08 | −0.31 | −0.07 | −0.21 | 0.03 | | | | | | | |
| Mg$^{2+}$ | −0.47 * | −0.23 | 0.03 | −0.22 | −0.06 | 0.33 | −0.14 | −0.16 | 0.35 | 0.84 ** | −0.01 | | | | | | |
| Na$^{+}$ | −0.46 * | 0.29 | −0.08 | −0.37 | 0.47 * | 0.10 | −0.38 | −0.30 | −0.32 | 0.10 | 0.71 ** | 0.21 | | | | | |
| Si | −0.33 | −0.13 | 0.31 | −0.08 | 0.35 | 0.06 | −0.13 | −0.02 | 0.09 | 0.59 ** | 0.33 | 0.40 | 0.51 * | | | | |
| Cl$^{-}$ | −0.53 * | −0.01 | −0.02 | −0.23 | 0.16 | 0.10 | −0.31 | −0.34 | −0.28 | 0.19 | 0.54 ** | 0.42 | 0.61 ** | 0.26 | | | |
| NO$_3^{-}$ | 0.24 | −0.17 | 0.15 | 0.00 | −0.08 | −0.18 | −0.23 | 0.12 | 0.58 ** | 0.57 ** | 0.30 | −0.26 | 0.28 | −0.30 | | | |
| SO$_4^{2-}$ | 0.17 | −0.02 | 0.04 | 0.04 | −0.13 | 0.04 | 0.14 | 0.12 | 0.67 ** | 0.70 ** | −0.06 | 0.54 ** | −0.18 | 0.06 | −0.10 | 0.39 | |
| HCO$_3^{-}$ | −0.51 * | −0.17 | 0.18 | −0.22 | 0.19 | 0.26 | −0.19 | −0.12 | 0.14 | 0.82 ** | 0.20 | 0.75 ** | 0.43 * | 0.78 ** | 0.439 * | 0.37 | 0.19 |

Note(s): * significant at $p < 0.05$ levels; ** significant at $p < 0.01$ levels.

**Table 5.** Principal component analysis matrix of heavy metals in DLQ.

| | PC1 | PC2 | PC3 |
|---|---|---|---|
| Cd | 0.297 | **0.606** | −0.100 |
| Cr | −0.049 | **0.643** | 0.143 |
| Mn | **0.624** | 0.457 | 0.108 |
| Fe | **0.902** | 0.273 | −0.104 |
| Ni | −0.256 | 0.251 | **0.866** |
| Cu | 0.261 | −0.235 | **0.864** |
| Zn | **0.906** | −0.148 | 0.052 |
| Pb | **0.866** | 0.397 | −0.038 |
| As | 0.185 | **0.558** | −0.074 |
| Eigenvalue | 3.46 | 1.56 | 1.25 |
| % of variance | 33.71 | 18.61 | 17.34 |
| Cumulative % | 33.71 | 52.32 | 69.66 |

Note(s): Bolded values are > 0.5.

PC1 accounted for 33.71% of the total variance and was mainly contributed by Fe, Zn, Pb, and Mn. These heavy metals were highly significantly correlated (Table 4) and had similar spatial distribution patterns (Figure 4); therefore, Fe, Mn, Pb, and Zn may have similar sources or chemical characteristics [25]. Generally, Fe and Mn are abundant in the earth's crust [12], and the background values of Pb and Zn in the watershed are relatively high [9,73]. Therefore, PC1 was mainly influenced by natural sources. In addition, Wang et al. [15] showed that the Pb and Zn concentrations in the surface soils were high in the major transportation routes of the Qinghai-Tibet Plateau, while the DLQ is close to the national highway G109 (Figure 1), suggesting that the concentrations of Pb and Zn in river water in the DLQ may be influenced by transportation. In general, the Pb contained in car exhaust enters the water through dry and wet deposition, which increases the concentration of Pb in the river water [15]. At the same time, Zn, Pb, and Mn particles are released by the abrasion of vehicle components and incomplete combustion of fuel [73]. Therefore, PC1 was mainly influenced by natural sources and, secondly, may be influenced by traffic activities.

PC2 accounted for 11.91% of the total variance and was mainly contributed by Cd, Cr, and As. Based on the spatial distribution of As in this study area (Figure 4), it could be observed that the high concentration of As mainly occurred after the inflow of geothermal water. The As concentration of river water in DLQ is primarily influenced by geothermal wastewater discharge [12], which is consistent with previous studies [23]. Therefore, PC2 was primarily related to the input of geothermal springs. In addition, the towns of Dongga and Naiqiong are important agricultural bases in the DLQ watershed, and pesticides and chemical fertilizers used in agricultural production are also important sources of As, Cd, and Cr [25]. Therefore, PC2 was mainly affected by geothermal activities and may be affected by agricultural activities secondarily.

PC3 accounted for 11.91% of the total variance and was mainly contributed by Ni and Cu. The LR watershed was rich in mineral resources (mainly Cu, Zn, and Pb ores) [9,41]. Studies have shown that Ni and Cu are influenced by the rich mineral resources and previously managed mining activities in the basin [12,41]. Therefore, PC3 was primarily affected by mining activities.

### 3.5.3. Source Apportionment Using APCS-MLR

Following identified potential sources, the contributions of heavy metals were quantified with the APCS-MLR. As shown in Table 6, the ratio of the mean observed and estimated values of heavy metals observed indicated a high credibility of the APCS-MLR analysis.

**Table 6.** Contribution of heavy metals pollution sources in DLQ.

| | PC1 | PC2 | PC3 | Unidentified Sources (%) | Estimated Mean Concentration (mg/L) | Observed Mean Concentration (mg/L) | Ratio (E/O) |
|---|---|---|---|---|---|---|---|
| Cd | — | 2.12% | — | 97.88% | 0.05 | 0.05 | 1.0000 |
| Cr | — | 93.07% | 6.93% | — | 1.65 | 1.65 | 1.0000 |
| Mn | 20.67% | 73.53% | 5.80% | — | 7.30 | 7.30 | 1.0000 |
| Fe | 40.46% | 59.54% | — | — | 78.36 | 78.36 | 1.0000 |
| Ni | — | 46.39% | 53.61% | — | 0.84 | 0.84 | 1.0000 |
| Cu | 15.64% | — | 84.36% | — | 1.03 | 1.03 | 1.0007 |
| Zn | 91.41% | — | 8.59% | — | 5.26 | 5.26 | 0.9996 |
| Pb | 26.15% | 58.28% | — | 15.57% | 0.53 | 0.53 | 0.9999 |
| As | 6.37% | 93.63% | — | — | 3.33 | 3.33 | 0.9995 |

Note(s): — The value is small and can be ignored.

The Cr, Mn, Fe, Pb, and As in DLQ were primarily derived from the influx of geothermal water, with contributions of 93.07%, 73.53%, 59.54, 58.28%, and 93.63%, respectively. This establishes that YBJ geothermal water strongly contributes to the heavy metals of river water in DLQ. In addition, Mn, Fe, and Pb were also influenced by natural factors, with 20.67%, 40.46%, and 26.15%, respectively. The contribution of natural factors was 91.41% for Zn. Ni mainly came from the combined influence of mining activities and geothermal springs, with contributions of 53.61% and 46.39%, respectively. Cu mainly came from mining activities, with a contribution of 91.41%. In addition, the existence of Cd and Pb in 97.88% and 15.57% of the unidentified sources needs to be further explored. In conclusion, natural factors, geothermal springs, and mining activities had different degrees of contribution to the nine heavy metals.

## 4. Practical Applications

The preliminary results of this work will help to fill the current knowledge gaps regarding the water environment in small watersheds on the TP. It will also provide information for the integrated and refined management of water ecosystems. In addition, this work provides a basis for the development of water environment risk contingency plans. Regular surveys of water quality around geothermal fields can reduce health risks to the residents in the surrounding area. Therefore, their implementation would be useful managing the environmental safety of the water environment in the future.

## 5. Conclusions

This study focuses on the composition of the major ions and the sources of heavy metals in the DLQ. The water ion chemistry in the LR and DLQ rivers is mainly produced by the weathering of carbonate rocks. In addition, solutes in the rivers are also affected by the inflow of geothermal water. heavy metals in the DLQ rivers are mainly from natural, geothermal, and mining sources. The source of heavy metals to the river from the inflow of geothermal water cannot be ignored. In addition, increased precipitation and runoff due

to climate change may have a diluting effect on the concentration of heavy metals such as As in rivers. The harsh environment of the highlands and the lack of meteorological and hydrological stations have led to a lack of long-term spatial and temporal observations of rivers. These factors limit the understanding of space-time variation of ion fractions and heavy metal pollution in small watershed rivers. Considering that the DLQ is used as a source of domestic drinking water, this study can provide a basis for adequate water quality protection and support further research on aquatic environmental evolution in small river basins in the Tibetan Plateau.

**Author Contributions:** W.X., methodology, writing, original draft preparation, and software; L.W., sample collection, methodology, and investigation; X.W. and W.M., sample collection and experiment; J.H., methodology and investigation; J.L. and X.L., conceptualization, investigation, data analysis, and editing manuscript. All authors have read and agreed to the published version of the manuscript.

**Funding:** This research was funded by the National Basic Research Program of China (2013CB956401) and National Natural Science Foundation of China (Nos. 41661144029 and 41672351).

**Institutional Review Board Statement:** Not applicable.

**Informed Consent Statement:** Not applicable.

**Data Availability Statement:** The datasets used or analyzed during the current study are available from the corresponding author on reasonable request.

**Conflicts of Interest:** The authors declare no conflict of interest.

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
