# Peer review of "Geochemistry and Sources Apportionment of Major Ions and Dissolved Heavy Metals in a Small Watershed on the Tibetan Plateau"

_water, doi:10.3390/w14233856_

Round 1

Reviewer 1 Report

The purpose of the manuscript “water_ 2021800” are to analyze the major ionic composition and water quality of Duilong Quo and identify the natural and/or anthropogenic sources of heavy metals using multivariate analysis techniques.

The paper appears well-structured but some sections must be improved or clarified.  Therefore, I believe the manuscript should be published only after major revision.

Specific comments:

R=939: Geogenic pollution is few o never mentioned in the introduction. I recommend reading the following works and integrating this part:

Bera, T., Kumar, V., Sarkar, D.J., Devi, M.S., Behera, B.K. and Das, B.K., 2022. Pollution assessment and mapping of potentially toxic elements (PTE) distribution in urban wastewater fed natural wetland, Kolkata, India. Environmental Science and Pollution Research, pp.1-20.

Fuoco, I., Marini, L., De Rosa, R., Figoli, A., Gabriele, B. and Apollaro, C., 2022. Use of reaction path modelling to investigate the evolution of water chemistry in shallow to deep crystalline aquifers with a special focus on fluoride. Science of The Total Environment, 830, p.154566.

R:150: a geological map, even schematic, should be included in the work.

R:152: Figure 1: it would be more appropriate to make this card in color to better distinguish the various symbols and other things illustrated

R:162: The Eh and alkalinity measurements are missing, which should also be done in situ!

R:202: use box plots to better describe and visualize these characteristics and variations.

R:241: To evaluate the chemical composition of the water it is not enough to use the Piper diagram because it does not take into account (as proposed by the authors) salinity, I suggest using a TIS salinity diagram, as proposed by:

Apollaro, C., Tripodi, V., Vespasiano, G., De Rosa, R., Dotsika, E., Fuoco, I., Critelli, S. and Muto, F., 2019. Chemical, isotopic and geotectonic relations of the warm and cold waters of the Galatro and Antonimina thermal areas, southern Calabria, Italy. Marine and Petroleum Geology, 109, pp.469-483.

R:292: anyway, it is necessary to refer to other processes that can lead to the enrichment of solafate and aresenic. Such as the dissolution of sulphides (pyrite) which are always present in crystalline rocks, as evidenced by the work of Fuoco et al 2022:

Fuoco, I., De Rosa, R., Barca, D., Figoli, A., Gabriele, B. and Apollaro, C., 2022. Arsenic polluted waters: Application of geochemical modelling as a tool to understand the release and fate of the pollutant in crystalline aquifers. Journal of Environmental Management, 301, p.113796.

R:361: describe the sorption processes. Referring to iron oxyhydrosides, as highlighted by Fuoco et al. 2022.

R:380: specify whether it is total chromium or hexavalent chromium, which is much more dangerous. And anyway, as evidenced by Fuoco et al 2020, almost always all the chromium in solution is always hexavalent:

Fuoco, I., Figoli, A., Criscuoli, A., Brozzo, G., De Rosa, R., Gabriele, B. and Apollaro, C., 2020. Geochemical modeling of chromium release in natural waters and treatment by RO/NF membrane processes. Chemosphere, 254, p.126696.

R:467: why were not the major components (cations and anions) included in the PCA calculation?

R:502: PCA could be better explained using correlation graphs between the main components, as evidenced for example by:

Gazley, M.F., Collins, K.S., Roberston, J., Hines, B.R., Fisher, L.A. and McFarlane, A., 2015. Application of principal component analysis and cluster analysis to mineral exploration and mine geology. In AusIMM New Zealand branch annual conference (Vol. 2015, pp. 131-139). Dunedin New Zealand.

Mohamed, A., Asmoay, A., Alshehri, F., Abdelrady, A. and Othman, A., 2022. Hydro-geochemical applications and multivariate analysis to assess the water–rock interaction in arid environments. Applied Sciences, 12(13), p.6340.

Discussions and conclusions need to be rewritten taking into account previous comments

Recommended works must be added in the bibliography

Bera, T., Kumar, V., Sarkar, D.J., Devi, M.S., Behera, B.K. and Das, B.K., 2022. Pollution assessment and mapping of potentially toxic elements (PTE) distribution in urban wastewater fed natural wetland, Kolkata, India. Environmental Science and Pollution Research, pp.1-20.

Fuoco, I., Marini, L., De Rosa, R., Figoli, A., Gabriele, B. and Apollaro, C., 2022. Use of reaction path modelling to investigate the evolution of water chemistry in shallow to deep crystalline aquifers with a special focus on fluoride. Science of The Total Environment, 830, p.154566.

Apollaro, C., Tripodi, V., Vespasiano, G., De Rosa, R., Dotsika, E., Fuoco, I., Critelli, S. and Muto, F., 2019. Chemical, isotopic and geotectonic relations of the warm and cold waters of the Galatro and Antonimina thermal areas, southern Calabria, Italy. Marine and Petroleum Geology, 109, pp.469-483.

Fuoco, I., De Rosa, R., Barca, D., Figoli, A., Gabriele, B. and Apollaro, C., 2022. Arsenic polluted waters: Application of geochemical modelling as a tool to understand the release and fate of the pollutant in crystalline aquifers. Journal of Environmental Management, 301, p.113796.

Fuoco, I., Figoli, A., Criscuoli, A., Brozzo, G., De Rosa, R., Gabriele, B. and Apollaro, C., 2020. Geochemical modeling of chromium release in natural waters and treatment by RO/NF membrane processes. Chemosphere, 254, p.126696.

Gazley, M.F., Collins, K.S., Roberston, J., Hines, B.R., Fisher, L.A. and McFarlane, A., 2015. Application of principal component analysis and cluster analysis to mineral exploration and mine geology. In AusIMM New Zealand branch annual conference (Vol. 2015, pp. 131-139). Dunedin New Zealand.

Mohamed, A., Asmoay, A., Alshehri, F., Abdelrady, A. and Othman, A., 2022. Hydro-geochemical applications and multivariate analysis to assess the water–rock interaction in arid environments. Applied Sciences, 12(13), p.6340.

Reviewer 2 Report

Dear authors, thank you for submitting the manuscript to the journal of Water. Its topic is very interesting. However, the current version of the paper suffers from a number of weaknesses related to the empirical strategy used. I have the following comments/questions for the authors:

Abstract

·         The abstract could be more specific. I suggest the authors should organize the abstract as well as main text in four sections, namely: scope, objectives, methods, results, conclusions. Also, abstract section should be completed with the results of the study. Don’t use abbreviation on the first time. Define full form for the first time than after use abbreviation only (Please check in the entire manuscript).

·         The abstract section needs a lot of improvement in scientific writing. There are many sentences which are not properly presented.

·         Add important results in the abstract section.

·         The authors ought to re-write the abstract so that it briefly presents the problem at hand, objectives of the study, methods used to achieve the objectives in logical order. Also, abstract section should be completed with the results of the study.

Introduction

·         In introduction chapter please focus on problem generally, on the basis of examples in the whole World, not your study area.

·         Add some facts and figures of surface water quality around the globe in your introduction.

·        Add some recent article to make your introduction more attractive and strong. I propose to add this survey method in the overview section of the introduction section, based on the latest literature. Please replace old citations (if it is possible) or add citations of newest literature.

https://doi.org/10.3390/w14071131

https://doi.org/10.1007/s12517-020-05882-x

Materials and Methods

Study area

·         Describe all the features of the study area in brief including climate, topography, geology, and hydrogeology?

·         Would you please give more information about the River (e.g., max depth, average precipitation and evaporation, the prevalent climate, mixing regime, warm monomictic or polymictic)?

Sampling and Analysis

·         Please give detailed information on water samplers (e.g., accuracy).

·         Sampling locations were selected carefully within Duilong Qu (DLQ), to have a good representation of the spatial variability of quality indicators across-section of water quality monitoring. What criteria where analyzed to select this locations?

·         Please provide detailed detection methods and quality control results?

·         Please support your methods by providing appropriate references or give the guidelines used to analyze the water quality parameters.

·         Please support your methods by providing appropriate references or give the guidelines used and equations.

·         How did you do quality control (QC) and quality assurance (QA) on the obtained data to validate the conclusions?

Results and Discussion

·         You should think how transformational the research is likely to be should be made so that the outcome of the work will have an impact on the community/society facing given sustainability related challenges?

·         Write the practical applications of your work in a separate section, before the conclusions and provide your good perspectives.

·         What are the bottlenecks of this work and how did you mitigate the impacts attributed to them?

·         What are the likely research impacts of this work globally, nationally and locally?

·         Make a table of comparison of this present work and other similar techniques from previously published study in terms of operational parameters; afterwards, please give a critical analysis on its technical feasibility and applicability for up scaling this process.

·         What long-term impacts will it have on environmental protection and the wider public or the field following the completion of the research?

Conclusion

·         Concise the text in conclusion and add future work in order to recommend your work. Shorten the length of each and every paragraph by adding only relevant and major findings in your study.

Please respond to all of those comments in the revised manuscript by pointing out precisely and concisely on which page and in which line you have incorporated your response one by one.

Reviewer 3 Report

The manuscript: Geochemistry and source distribution of major ions and dissolved heavy metals in a small watershed on the Tibetan Plateau, presents a case study that investigated the variations of major ions and quantified the sources of heavy metals in the Duilong Qu. This work is one of the best studies that I have seen in this area, However, a few remarks should be noted:

ü The samples were collected in 2015, why were these results not published before? For this, You must mention in the title, summary and conclusion that the samples were collected in 2015 and give the reasons.

ü In figure 5 you need to add the references of this data

ü You must add to tables 1 and 2 the WOH limit for different elements.

In conclusion, I recommend this manuscript for publication with minor corrections.

Round 2

Reviewer 1 Report

Remarks from reviewer have been correctly addressed, and the paper is now more focuse on his core topic
In my opinion it is now acceptable.
Best regards

Reviewer 2 Report

This paper is an interesting study and authors have investigated the major ions and heavy metals of river water in a small watershed on the Tibetan Plateau to identify the natural and/or anthropogenic sources of heavy metals, and the effect of geothermal hot springs and climate change on concentrations of river water in in the Duilong Qu (DLQ).

The article is written correctly, includes a discussion of the research findings, and a good review of the literature. The results are presented in a clearly structured manner. The manuscript has been significantly improved and can now be accepted in current form.
